# The Influence of Cannabinoids on Multiple Myeloma Cells: A Scoping Review

Karan Varshney [1,2,*] , Prerana Ghosh [1] and Akash Patel [3]

1 School of Medicine, Deakin University, Victoria, VIC 3216, Australia
2 School of Public Health and Preventive Medicine, Monash University, Melbourne, VIC 3004, Australia
3 Midwest Dental Canton, Canton, IL 61520, USA
* Correspondence: karan.varshney@students.jefferson.edu; Tel.: +1-604-359-6721

**Abstract:** Multiple myeloma (MM) is one of the most common hematological malignancies. There is a clear need for research into new treatment options that can improve the life expectancy and quality of life for MM patients; this is particularly salient for those with relapsed/refractory disease. Cannabinoids (CB) have shown potential in treatment regimens for a number of cancers, but little is currently known about their effectiveness against MM. Hence, we conducted a scoping review regarding the usage of CB against MM cells. For our review, searches were conducted in PubMed, Web of Science, and OVID Medline. After screening, six articles were eligible for inclusion, all of which were laboratory studies. It was demonstrated that CB decrease MM cell viability, and this was consistently shown to occur alongside the activation of apoptotic pathways in MM cells. These effects were shown to continue to occur in dexamethasone-resistant MM cells. The effects of CB on MM cells were enhanced when used in combination with standard treatments for MM. Critically, these marked decreases in MM cell viability induced by CB did not occur in non-MM cells. Overall, these findings indicate a clear need for future clinical trials of the integration of CB into MM treatment regimens.

**Keywords:** multiple myeloma; cannabinoids; tetrahydrocannabinol; treatment; chemotherapy; oncology





## 1. Introduction

Multiple myeloma (MM) is a hematological malignancy of B cells occurring when genetically mutated plasma cells fail to undergo cell apoptosis and accumulate in the bone marrow [1]. Generally, a diagnosis of MM consists of two components: the clinical presentation of anemia, hypercalcemia, renal disorder, and bone lesions; and a bone marrow biopsy confirming a plasma cell population of greater than or equal to 10% [2]. In contrast to normal bone remodeling, the coupling mechanism of osteoclasts and osteoblasts is lost in MM. Increased osteoclastic activity, resulting in bone resorption, and suppressed osteoblastic activity, leading to decreased/absent bone formation, are key factors in the development of bone destruction in MM [2].

MM is the second most common hematological malignancy, with nearly 35,000 cases projected to be diagnosed in 2022 across the United States; rates are higher in men, older individuals, and individuals of African descent [3–8]. Survival rates among patients with MM have been consistently improving over time since the 1990s. From 2000 to 2008, the five-year survival rate was found to be between 34.5% and 49.6% [9], and increased to 53.9% between 2010 to 2016 [10]. The explanation for this rise in survival rates among MM patients is the increased availability of new treatments for this disease.

Previously, melphalan-based treatment regimens were primarily utilized in MM treatment for roughly forty years. This was until the introduction of thalidomide, lenalidomide (immunomodulatory therapies), bortezomib (proteasome inhibitor), and stem cell transplantation in the 1990s [3,11–18]. While there currently are a number of different treatment regimens, the combination of bortezomib, lenalidomide, and dexamethasone has become

the standard of treatment for patients with MM; this combination treatment has demonstrated very high response rates and has been shown to result in patients emerging from treatment with complete response/very good partial response [19,20].

While five-year survival rates have improved in recent years for MM patients, there is nonetheless a very clear need for the development of new treatments. Although MM is treatable, it remains an incurable disease. In cases of relapsed/refractory disease, treatment options remain fairly limited—one emerging form of treatment that has shown emerging potential in very recent times is B cell maturation antigen (BMCA)-targeted approaches, which may utilize options such as antibody–drug conjugates (ADCs), bispecific T cell engagers (BITEs), and chimeric antigen receptor (CAR) T cell therapy [21]. While promising, these forms of treatment have been shown to cost hundreds of thousands of dollars [22,23]. Treatment options also become limited in the case of drug-resistant disease. Furthermore, despite the progress in survival rates for MM, five-year and ten-year mortality rates are still notably higher, particularly in older individuals [24,25]. There is therefore a clear need for newer, cost-effective treatment options for MM.

One such treatment that may offer potential for MM patients is cannabinoids (CB). CB have been shown to have antitumor effects in a number of different cancers, such as colorectal cancer, as well as cancer of the lung, brain, prostate, and breast [26,27]. Notably, it has also been proposed that CB may have a positive health impact for those with cancer cachexia by improving low appetite, which is an issue for some MM patients [28]. CB may also have a role in improving a number of other aspects of quality of life for cancer patients by addressing issues such as anxiety, pain, vomiting, and other related issues [29,30]. Figure 1 summarizes the means by which CB have been proposed to improve quality of life and life expectancy for patients with cancers such as MM.

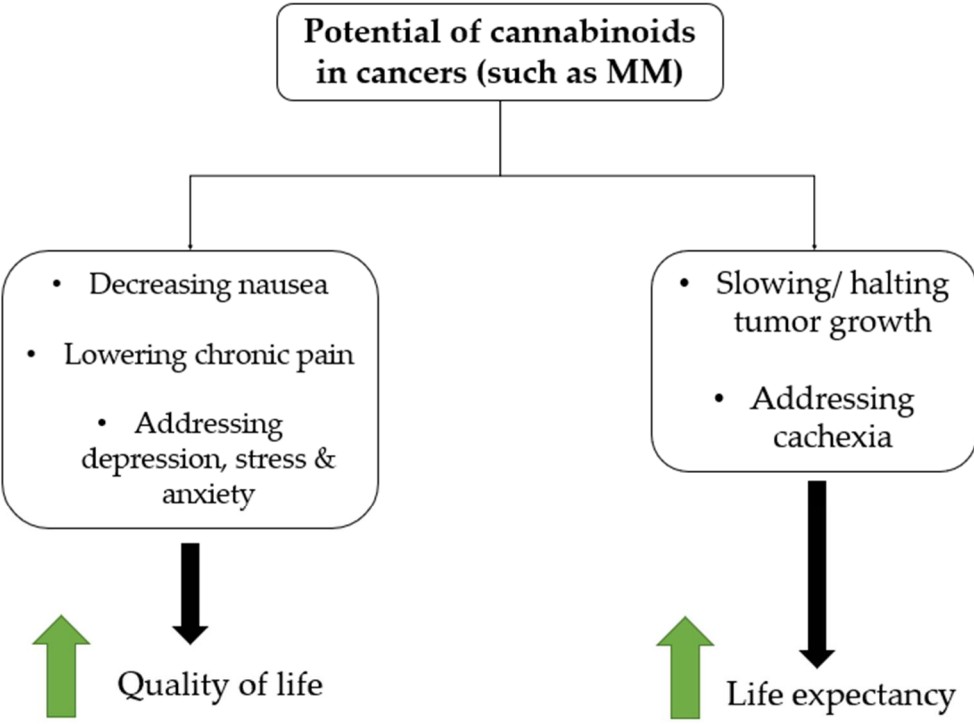

**Figure 1.** Hypothesized therapeutic potential of cannabinoids in cancers such as MM.

While the potential of CB is increasingly being understood for a wide range of malignancies, many previous articles and reviews have focused very generally on the therapeutic role of CB in various cancers. For example, three reviews on the role of CB in the treatment of cancer placed minimal focus on any particular cancer and instead focused on a large array of different cancers [29–31]; as a result, little is currently known regarding the effects of CB on MM and the therapeutic potential of this compound for MM patients.

The limited understanding of the role of CB in MM treatment from previous reviews has important implications. First of all, the limited understanding of the risks and benefits may lead to clinicians withholding a recommendation of CB for MM patients or potentially making misinformed recommendations based on findings relating to other cancers. Second, it is an imperative to clear up misconceptions for cancers such as MM, as misinformation regarding CB treatment has been shown to be widespread in public platforms [32]. Therefore, in order to provide a clearer understanding of the role of CB in MM, the overall aim of this work is to provide a scoping review of the literature regarding the influence of CB on MM cells and MM patient outcomes.

## 2. Materials and Methods

This scoping review followed the 'Preferred Items for Systematic Reviews and Meta-Analysis extension for Scoping Review' (PRISMA-ScR) guidelines [33,34]. On 23 July 2022, searches were conducted in PubMed, OVID Medline, and Web of Science. Searches included terms for CB and MM, and MeSH terms, or their equivalent, were used. For the searches, no restrictions were placed based on date of publication. Table 1 lists the search terms used for OVID Medline.

**Table 1.** Search terms for OVID Medline (conducted on 23 July 2022).

| | |
|---|---|
| **Search terms:** ((cannabinoids.mp) OR (Dronabinol.mp) OR (Cannabidiol.mp) OR (cannabis.mp) OR (Cannabaceae.mp) OR (THC) OR (delta-9-tetrahydrocannabinol) OR (cannabinoid) OR (canabinoid)) AND ((Multiple Myeloma.mp) OR (Myeloma Proteins.mp) OR (myelomas) OR (myeloma) OR (myelomatos) OR (Kahler disease)) | **Result total: 18** |

After retrieval from the respective databases, two experienced reviewers (KV and PG) screened articles independently. First, duplicate articles were removed. Next, all papers were screened by title/abstract. Thereafter, the full texts of all remaining articles were screened and analyzed in order to determine eligibility for inclusion in this review. The inclusion criteria were broad in order to include as many relevant articles as possible; articles were included if they met the following criteria:

- Quantitative, original research;
- Was a peer-reviewed, full-text article;
- Written in English;
- Tests the effects of CB treatment/CB receptor upregulation in MM patients/MM cell lines.

For this review, data were extracted regarding the study characteristics, the details of the study methods, and the key findings of the article. More precisely, the following data were extracted: year, country, aim, drug under evaluation, cell types/receptors being studied/influenced, analyses, and main findings. After the extraction of data was completed, relevant data were qualitatively synthesized, with patterns and trends being described.

## 3. Results

### 3.1. Searches and Included Articles

The searches produced a total of 58 results. After removal of 30 duplicates, 28 articles remained. Once articles were screened by title/abstract, a total of 9 articles remained for full-text analysis; 6 articles were ultimately eligible for inclusion in the review [35–40]. The full workflow for the screening of the articles are shown in Figure 2.

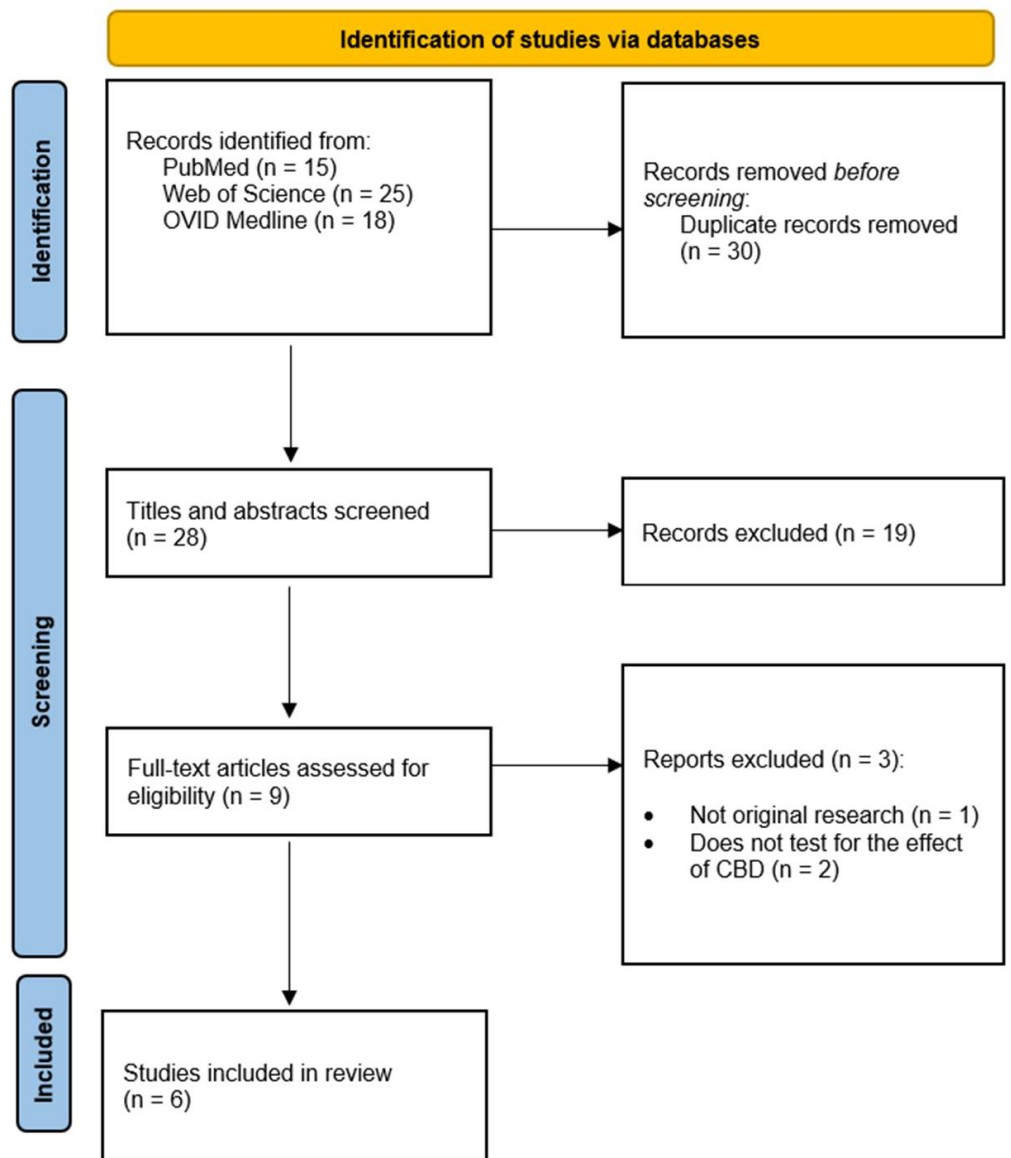

**Figure 2.** Process of screening articles based on the PRISMA 2020 flow diagram [34].

### 3.2. Study Characteristics

The studies included in this review were conducted in the following countries: USA (*n* = 1), Germany (*n* = 1), Qatar (*n* = 1), Spain (*n* = 1), and Italy (*n* = 2). Studies were conducted between 2013 and 2021. There were no clinical trials eligible for this review; all included articles were laboratory studies. All six studies had in vitro/ex vivo analysis, although one article included both in vitro and in vivo analysis.

The following drug compounds were used in differing studies: WIN-55, PGN6, PGN17, PGN34, PGN72 [35], phenylacetamide (PAM) [36], pure cannabidiol (CBD) [37,39], B-caryophyllene (BCP) [38], and tetrahydrocannabinol (THC) [40]. Alongside these substances, the following treatments were used in combination with substances containing/activating CB: dexamethasone [35], melaphalan [35], bortezomib (BORT) [39], and carfilzomib (CFZ) [40]. CB2 was the most common receptor analyzed.

Complete study and drug characteristics for all included studies are listed in Table 2.

**Table 2.** Study and drug characteristics from included studies.

| Author | Country | Aim | Drug | Cell Type/Receptor | Analyses | Main Findings |
|---|---|---|---|---|---|---|
| Barbado et al. (2017) [35] | Spain | To study the effects of certain cannabinoids on proliferation and viability of myeloma plasma cells in vitro, as well as in vivo. | WIN-55, PGN-6, PGN-17, PGN-34, PGN-72 | CB2 | Assessed the effect of CB on cell viability using MTT assay and flow cytometry, Western blot analysis of CB-induced apoptotic mechanisms, protein expression and molecular pathways, confirmed the effects of CB mediated by CB2R, analyzed the effect of CB on other anti-myeloma agents using MTT assay and investigated the antitumor effect of CB in vivo using xenograft models. | PGN cannabinoids significantly ↓ MM cell viability. ↑ in pro-apoptotic proteins Bak and Bax, and ↓ expression of anti-apoptotic proteins Bcl-xL and Mcl-1. The apoptotic caspase-2 pathway was the most strongly activated. Akt is most strongly modulated, with a biphasic response. Ceramide was shown to have a major role in cannabinoid-induced apoptosis of MM cells. Slight, but sustained, ↓ in ER-stress protein markers such as CHOP, ATF-4, p-IRE1, and XBP-1 sec. WIN-55 has a synergistic effect in combination with dexamethasone and melphalan and overcame melphalan-resistance. CB administration ↓ tumor-volume. |
| Feng et al. (2015) [36] | USA | To provide insights regarding how CB2 ligands exert anti-MM effects and provide rationale for future in vivo investigations. | PAM | CB2 | Confirmation of significance of CB2R pathway in PAM-induced myeloma cell apoptosis with the use of gene silencing, H-thymidine incorporation assay, computer molecular modeling and docking studies, assessment of apoptotic cell death and cell viability, cell cycle analysis with flow cytometry, RT-PCR, and Western blotting. | PAM behaves as an inverse agonist of CB2R, and PAM may dock to the binding pocket of CB2 inverse agonist SR114528. PAM has antiproliferative and growth-inhibitory effects on myeloma cells. MM cells resistant to dexamethasone and melphalan exhibited either similar, or better, responses to PAM treatment, indicating that PAM may be able to overcome chemoresistance. PAM-induced apoptosis involves both caspase-independent and caspase-dependent pathways. It also ↓ survivin levels. PAM may exert its negative regulation of cell cycle of MM cells at different levels in the checkpoint. PAM is suggested to have an influence on cytoskeletal proteins via posttranslational modification mechanisms. |

**Table 2.** *Cont.*

| Author | Country | Aim | Drug | Cell Type/Receptor | Analyses | Main Findings |
|---|---|---|---|---|---|---|
| Garofano & Schmidt-Wolf (2020) [37] | Germany | To determine the antitumor effect of CIK cells combined with pure CBD in KMS-12 MM cells. | Pure CBD | CB2, CIK cells | Flow cytometry analysis of expression of CB2R and NKG2D in cells, analyzed the effect of CBD alone and in combination with CIK cells on MM cell viability and cytotoxicity using CCK8, LDH assay and flow-cytometry analysis, determined effect of CBD on NKT cell growth using flow cytometry. | CBD significantly ↓ LDH release in CIKs. CBD significantly ↑ LDH release in KMS-12 PE cells. CBD significantly ↓ cytotoxic activity of CIKs against MM cells at high concentrations. CBD significantly ↑ absolute number of alive CIK cells relative to control at 1 μM, 3 μM, 5 μM and 10 μM concentration and significant ↓ at 20 μM. CBD↓ the percentage of NKT cells relative to controls. |
| Mannino et al. (2021) [38] | Qatar | To evaluate the antiproliferative and anticancer effects of BCP for MM.1R and MM.1S cells. | BCP | CB2 | After treating the cells, utilized FDA/PI staining for determining cell viability, MTT assay to evaluate cancer cell viability after BCP treatment, Trypan blue dye to quantify number of living and dead cells, ELISA to determine CDK4, CDK6, and Wnt1 levels, Western blot analysis to determine protein expression levels, and immunofluorescence staining. ANOVA with Tukey's post hoc test for statistical analysis. | BCP selectively ↓ cell viability in MM cell lines but not healthy control populations. BCP significantly ↑ number of non-viable MM cells and ↓ number of viable MM cells. BCP induced apoptotic process in MM cells, by ↑ in expression of caspase-3, Bax, and ↓ Bcl-2 expression. BCP had anti-proliferative effects by significantly ↓ Wnt1 levels, p-AkT and $\beta$-catenin protein expression. BCP significantly ↓ CDK4 and CDK6 levels and cyclin D1 protein expression. |
| Morelli et al. (2013) [39] | Italy | To evaluate the expression of TRPV2 in MM cells, and the TRPV-2 independent and dependent effects of CBD with BORT and CBD alone in MM cells. | CBD | CD138, CD34, TRPV2 | Cells were isolated, and compounds were obtained. The following analyses were conducted: FISH analysis, FACS analysis, Western blot analysis, flow cytometry analysis, colony forming assay, gene expression analysis, MTT assay, BrdU cell proliferation assay, cell cycle analysis, apoptosis assay, JC-1 staining, fluorescent probe DCFDA, DNA fragmentation assay, ELISA assay. ANOVA or Student's *t*-test used for statistical analysis. | ↑ susceptibility to CBD in TRPV2+ MM cells compared to TRPV− MM cells. CBD and BORT combination treatment had a synergistic effect on MM cell viability with no effect on CD34+ cell growth. BORT and CBD combination treatment strongly ↓ cell proliferation and arrested cell cycle at G1. CBD and BORT significantly ↑ mitochondrial and ROS-dependent necrosis in MM cell lines. CBD and BORT synergized to ↓ pERK levels and inhibited/abrogated both ERK activation and AKT phosphorylation. CBD alone, and in combination with BORT, ↓ DNA binding activation of p52, p65, and RelB NF-kB subunits. |

**Table 2.** *Cont.*

| Author | Country | Aim | Drug | Cell Type/Receptor | Analyses | Main Findings |
|---|---|---|---|---|---|---|
| Nabissi et al. (2016) [40] | Italy | To evaluate the effects of THC alone, and in combination with CBD, on MM cell lines, and the effect of both of these in combination with CFZ. | THC, CBD | CB2, CXCR4 | MM cell lines, THC, and CBD compounds were obtained. Thereafter, the following analyses were conducted: MTT assay, cell cycle analysis, apoptosis assay, PI-staining, Western blot analysis, DNA fragmentation assay, RT-PCR analysis, cell migration assay. ANOVA or Student's *t*-test were used for statistical analysis. | THC alone and in combination with CBD ↑ cytotoxicity of MM cells in a CB2R independent manner. THC-CBD combination was statistically more effective at arresting cells at G1 phase of cell cycle and ↑ cell accumulation in G1 and sub-G1 phases. THC-CBD combination induces autophagic cell death in MM cells. CBD alone ↑ LC3-II/LC3-I ratio, and THC-CBD greatly ↑ levels of LC3-II and LC3-II/LC3-I ratio; combination treatment also ↓ p62 levels. THC-CBD combination ↑ necrotic cell death and ↑ levels of H2AX compared to single treatments. THC-CBD ↓ the increased expression of β5i in MM cells and impaired expression of mature and precursor forms. THC-CBD synergizes with CFZ to significantly ↓ MM cell viability and induce cytotoxic effects. CFZ-THC-CBD cotreatment significantly ↓ CXCR4 and CD147 mRNA expression and ↓ SDF-1- and eCyPa-mediated chemotaxis in MM cells. |

Abbreviations. ↓: decrease. ↑: increase. BCP: Beta-caryophyllene. BORT: Bortezomib. β5i: Beta type-5 (subunit of immuno-proteosome). CB: Cannabinoid. CB2R: Cannabinoid receptor-2. CBD: Cannabidiol. CDK: Cyclin-dependent kinase. CFZ: Carfilzomib. CHOP: C/EBP homologous protein. CIK: Cytokine-induced killer cells. H2AX: Phosphorylated variant of histone 2A. LC3-I: Soluble form of microtubule-associated protein light chain 3. LC3-II: Lipidated and autophagosome-associated form of light chain 3. LDH: Lactate dehydrogenase. MM: Multiple myeloma. MM.1R: Dexamethasone-resistant human MM cells. MM.1S: Dexamethasone-sensitive human. MM cells. NKT: Natural killer T cell. PAM: Phenylacetylamide. THC: $\Delta^9$-Tetrahydrocannabinol. TRPV2: Transient receptor potential vanilloid type-2 channel. WIN-55: WIN-55,212-2 mesylate.

### 3.3. Effects of CB on MM cells

The overall effects of CB on MM cells are summarized and depicted in Figure 3.

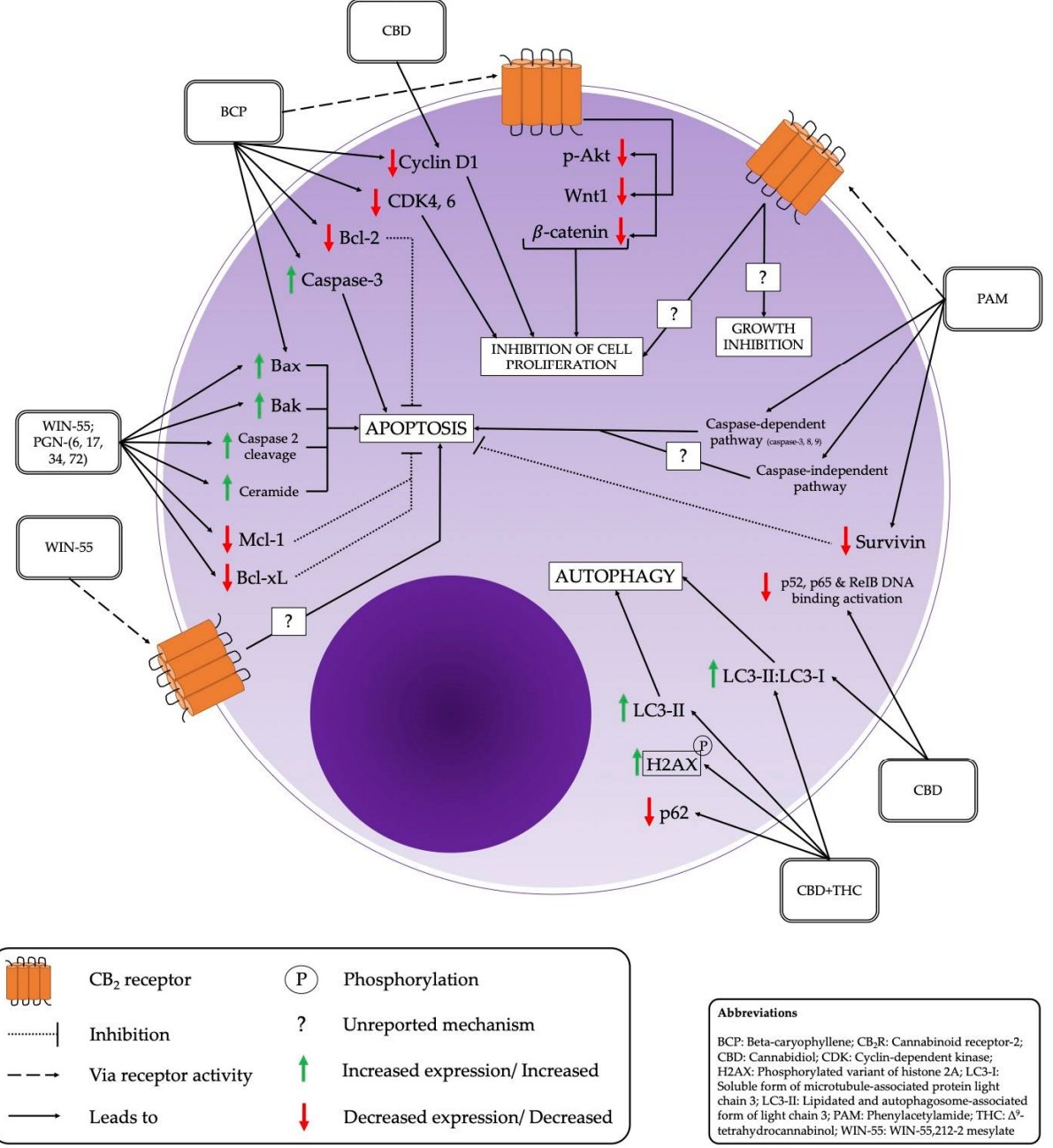

**Figure 3.** Effects of cannabinoids on multiple myeloma cells and associated pathways.

### 3.3.1. Cell Viability

MM cell viability was decreased consistently across all studies after the administration of CB/CB-containing compounds. This effect was demonstrated to occur in a dose-dependent manner. For example, KMS-12 PE cells (MM cells) exposed to CBD at 1–20 µM for 24 h had significant decreases in cell viability compared to controls [37]. Similarly, BCP at 50 µM resulted in a significant reduction in the number of viable MM cells, alongside a significant increase in non-viable MM cells [38]. Notably, in the only in vivo analysis in these studies, CB was shown to result in a major, progressive decrease in tumor volume [35].

### 3.3.2. Apoptotic Effects/Pathways

Across studies, apoptotic effects were also seen consistently. PGN cannabinoids showed a selective pro-apoptotic effect in MM cells [35], as did CBD and BCP administration respectively [38,40]. A number of different apoptotic pathways were activated, depending on the type of drug utilized. Table 3 lists the apoptotic pathways/effects associated with respective compounds.

**Table 3.** Compounds/drugs and corresponding apoptotic pathways/effects.

| Compounds/Drugs | Apoptotic Pathways/Effects |
|---|---|
| WIN-55, PGN-(6, 17, 34, 72) [35] | Increase in pro-apoptotic proteins Bak and Bax. Decreased expression of anti-apoptotic proteins Bcl-xL and Mcl-1. The apoptotic caspase-2 pathway most strongly activated. Ceramide shown to have a major role in CB-induced apoptosis |
| PAM [36] | Both caspase-independent and caspase-dependent pathways activated. Lower survivin levels (anti-apoptotic protein). |
| CBD [37,39] | LDH release significantly higher than in controls [37]. Mitochondrial and ROS-dependent necrosis [39]. |
| BCP [38] | Induced an increase in caspase-3 and Bax (apoptotic), and decrease in Bcl-2 expression (anti-apoptotic). Showed a significant reduction in p-AkT, Wnt1, and B-catenin–thereby demonstrating an anti-proliferative effect. |
| THC and CBD [40] | Shown to induce a minor increase in LC3-II/LC3-I ratio. |

### 3.3.3. Combinatory Effects

It was consistently demonstrated that the combinatory effect of CB with other drugs that are part of the standard treatment for MM is stronger than the effect of using either drug alone. For example, WIN-55 in combination with dexamethasone and melphalan showed a stronger anti-MM effect than utilization of any of these substances alone [35]. Similarly, use of BORT and CBD had a synergistic effect in inhibiting MM cells than the use of BORT only or CBD only [39]. THC along with CBD had a stronger effect in combination, and the potency of this anti-MM effect was further amplified when CFZ was used alongside the THC–CBD combination [40].

### 3.3.4. Effects on Drug-Resistant Cells

Numerous studies demonstrated that the use of CB/activation of CB receptors was capable of overcoming drug-resistance in MM. PAM treatment on cells resistant to dexamethasone and melphalan exhibited effects that were comparable to drug-sensitive MM cells [36]. Treatment of 50 µM BCP was shown to reduce the viability of dexamethasone-resistant MM cells to 80%, and 100 µM treatment to 50% [38]. The combinatory effect of WIN-55 with dexamethasone and melphalan was shown to be capable of overcoming melphalan-resistance [35].

### 3.4. Effects of CB on Normal Cells

While the effect of CB was consistently shown to reduce the viability of MM cells, CB was shown to have either a minor effect or no detrimental effect on normal (non-MM) cells. While PAM usage did lead to a decrease in MM cell viability, the cytotoxic effects on normal mononuclear cells were minor [36]. PGN cannabinoids showed a pro-apoptotic effect that occurred in MM plasma cells, but this did not occur in normal cells, including

hematopoietic stem cells [35]. Furthermore, the usage of BCP affected cancer cell viability, but did not have an effect on normal cells [38].

## 4. Discussion

Overall, the findings of this review show that there is consistent evidence that CB demonstrate a pro-apoptotic effect that reduces the viability of MM cells. This can occur by an array of different apoptotic pathways/effects, depending on the type of compound. These effects are amplified when used in combination with standard MM treatment drugs, such as dexamethasone, melphalan, and CFZ, and the effects show potential to overcoming MM drug resistance. Critically, these cytotoxic effects are not shown to occur in normal cells.

It must be emphasized that all the articles in this review were laboratory studies; due to the lack of clinical data regarding the usage of CB in MM treatment, clinical recommendations to offer CB cannot be made at this point. Nonetheless, the fact that CB showed an antitumor effect in vivo [35] highlights that these results offer enormous potential. Therefore, these findings demonstrate a clear need for preclinical and clinical trials to be conducted to test the effectiveness and safety of integrating CB in MM treatment regimens. This will be particularly valuable for MM patients with relapsed/refractory or drug-resistant disease who are frequently left with few treatment regimens that are affordable. Furthermore, this will provide far more insights into the other benefits and risks of CB for MM patients relating to appetite and quality of life. As MM patients have been shown to describe financial issues as a serious impact of MM and treatment [41], the low cost of CB offers additional promise for MM patients.

There are likely some ethical concerns regarding the usage of CB in such clinical trials. However, conducting clinical trials for MM patients with the integration of CB can occur in a safe manner. Control groups can be provided with standard treatment regimens, and experimental groups can be provided with the exact same treatment regimens along with the addition of CB in the form of WIN-55, PGN-6, PGN-17, PGN-34, PGN-72, PAM, or BCP. While one such trial was initially proposed back in 2018, recruitment for the trial never occurred [42]. The fact that CB were shown to have synergistic effects with standard MM treatment drugs, and that they have been shown to have minimal/no effects on normal cells, highlights that the integration of CB in trials can potentially be performed in a manner that is safe for patients.

The findings of this review also emphasize a clear need for further study on the tests of CB on MM, as well as for other cancers. The consistency of the effects shown in this review, and in studies on other cancers [26,27], demonstrates the importance of developing a deeper understanding regarding the pathways of biological effects that CB have on tumors. These findings also need to be conducted regarding cancer cachexia, which, to date, has very limited treatment options [43].

It is worthwhile to note that this review may be able to clear up misinformation regarding CB in MM—while some potential has been shown in the included laboratory studies, no definitive statements can be made at this time regarding the safety and effectiveness of the usage of CB in treatment regimens for MM patients. Such recommendations cannot be made until clinical trials are completed, again emphasizing the imperative for such trials to be conducted.

The findings of this review must be considered alongside the limitations. Only six studies were included, which is a relatively low number. It is possible that including more, and larger scale studies, may have revealed differing findings. Furthermore, as already alluded to, there were no clinical studies included. Therefore, clinical guidelines and recommendations cannot be made at this point regarding the effect of CB on MM patients. An additional limitation is that, while all the drugs/compounds included had CB/had effects on CB receptors, there was little consistency in the types of drugs/compounds studied. This was also true for the combinatory drugs. This limits the extent to which results can be generalized. Regardless of these limitations, it is imperative to denote that, across all six studies, CB showed very consistent effects on MM cell viability and consistent

cytotoxic effects on MM cells. These findings will be vital in guiding future studies that can potentially lead to changes that can improve the quality of life of MM patients.

## 5. Conclusions

Our scoping review has shown that CB may be highly effective for MM treatment due to their antitumor effect. While there is a clear need to study these effects in far more detail and for large-scale clinical trials in the future, these findings provide important first steps in potentially improving the management of MM.

**Author Contributions:** Conceptualization: K.V., P.G., A.P.; methodology: K.V., P.G.; validation: K.V., P.G.; formal analysis: K.V., P.G.; data curation: K.V., P.G.; writing—original draft preparation: K.V., A.P.; writing—review and editing: K.V., P.G., A.P.; visualization: K.V. All authors have read and agreed to the published version of the manuscript.

**Funding:** This research received no external funding.

**Institutional Review Board Statement:** Not applicable.

**Informed Consent Statement:** Not applicable.

**Data Availability Statement:** Not applicable.

**Acknowledgments:** The authors have no acknowledgements to be made.

**Conflicts of Interest:** The authors declare no conflict of interest.

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
