# Peer review of "The Influence of Cannabinoids on Multiple Myeloma Cells: A Scoping Review"

_futurepharmacol, doi:10.3390/futurepharmacol2030024_

Round 1

Reviewer 1 Report

The short review was clear, however, I suggest that in the introduction some information about the new therapies for MM (anti-BCMA and CAR-T) should be described.

Reviewer 2 Report

The review is about the use of cannabinoids in multiple myeloma.

The structure of the text is unusual. Being a review it should not be written as a research article, especially since there is not a great deal of data to discuss. A review is a critical article regarding specific topic. Here the authors have simply put together the results of 6 articles.

Especially the authors have a great confusion between cannabinoids in general and cannabidiol (CBD). The abbreviation CBD should not be used to identify cannabinoids in general.

Lines 28-31 should  be moved after line 34.

Ref 4 and 5 should be updated because authors are talking about 2022.

There are various mistakes to be corrected even typing

Reviewer 3 Report

In this manuscript, the authors deal with The Influence of Cannabinoids on Multiple Myeloma Cells

The topic is of interest and I have the next comments for improvement of the MS:

Introduction: add more data regarding MM and the correlation with cannabinoids. Also, a cartoon will be welcomed by the readers, for a better understanding

Material methods: all the terms used for searching should be MeSH terms. Revise it.

Table 2: this contains too many acronyms, and all abbreviations mentioned must be explained below the table. Also, there is too much text included. You can use arrows:↓ or ↑ instead of decrease, and decrease respectively.

The molecular mechanisms of the CBD’s effects on MM cells must be summarized in a figure.

There are a lot of published papers on this topic. What is the novelty of this paper?

What perspectives for human health does this MS have?

Consider revision accordingly.

Round 2

Reviewer 3 Report

The authors revised the MS accordingly. Only one minor point before the final approval: regarding Figure 3, I recommend adding a legend with all symbols and acronyms below the figure.

Author Response

Reviwer comment: The authors revised the MS accordingly. Only one minor point before the final approval: regarding Figure 3, I recommend adding a legend with all symbols and acronyms below the figure.

Response: The authors appreciate the recommendation made regarding Figure 3 and agree that a complete legend would enhance clarity. A completed figure legend with symbols has been added and all abbreviations have been appropriately defined below the figure. Additionally, minor changes were made to the pathways demonstrated in Figure 3 to make it more informative for readers.